# Physiological oxygen concentration during sympathetic primary neuron culture improves neuronal health and reduces HSV-1 reactivation

Sara A. Dochnal,[1] Patryk A. Krakowiak,[1] Abigail L. Whitford,[1] Anna R. Cliffe[1]

**ABSTRACT** Herpes simplex virus-1 (HSV-1) establishes a latent infection in peripheral neurons and periodically reactivates in response to a stimulus to permit transmission. *In vitro* models using primary neurons are invaluable to studying latent infection because they use bona fide neurons that have undergone differentiation and maturation *in vivo*. However, culture conditions *in vitro* should remain as close to those *in vivo* as possible. This is especially important when considering minimizing cell stress, as it is a well-known trigger of HSV reactivation. We recently developed an HSV-1 model system that requires neurons to be cultured for extended lengths of time. Therefore, we sought to refine culture conditions to optimize neuronal health and minimize secondary effects on latency and reactivation. Here, we demonstrate that culturing primary neurons under conditions closer to physiological oxygen concentrations (5% oxygen) results in cultures with features consistent with reduced stress. Furthermore, culture in these lower oxygen conditions diminishes the progression to full HSV-1 reactivation despite minimal impacts on latency establishment and earlier stages of HSV-1 reactivation. We anticipate that our findings will be useful for the broader microbiology community as they highlight the importance of considering physiological oxygen concentration in studying host-pathogen interactions.

**IMPORTANCE** Establishing models to investigate host-pathogen interactions requires mimicking physiological conditions as closely as possible. One consideration is the oxygen concentration used for *in vitro* tissue culture experiments. Standard incubators do not regulate oxygen levels, exposing cells to oxygen concentrations of approximately 18%. However, cells within the body are exposed to much lower oxygen concentrations, with physiological oxygen concentrations in the brain being 0.55%–8% oxygen. Here, we describe a model for herpes simplex virus 1 (HSV-1) latent infection using neurons cultured in 5% oxygen. We show that culturing neurons in more physiological oxygen concentrations improves neuronal health to permit long-term studies of virus-cell interactions and the impact on reactivation.

**KEYWORDS** herpes simplex virus, oxygen, *in vitro* culture, latency, reactivation, sympathetic neurons

Herpes simplex virus-1 (HSV-1) establishes a latent infection in peripheral neurons and reactivates to cause disease. *In vitro* model systems are beneficial for studying the molecular mechanisms of HSV-1 latent infection because neurons in culture can be more readily manipulated. Modeling HSV-1 latency and reactivation is technically challenging and requires the use of primary or differentiated neurons, which are post-mitotic and therefore do not replenish. Neurons also require specialized coating for adherence and supplements for survival and maturation. One challenge with working

Address correspondence to Sara A. Dochnal, sdochnal@health.ucsd.edu.

The authors declare no conflict of interest.

See the funding table on p. 7.

10.1128/spectrum.02031-24 **1**

with any cell type *in vitro* is keeping culture conditions as consistent as possible with their *in vivo* environment.

HSV-1, like many herpesviruses, is triggered to reactivate in response to cellular stress. Therefore, unwanted stress during cell culture may lead to inadvertent reactivation events and confounding experimental results. We recently developed a model system in which a period of 30 days is required for full entry into a latent infection (1). Culturing primary neurons for this length of time can be problematic as it can be challenging to keep primary neurons healthy. Therefore, we sought to optimize neuronal health to enable culturing neurons for this extensive time frame and minimize inadvertent stress and impacts on latency and reactivation.

Although widely accepted in scientific research, the conventional use of incubators that do not regulate oxygen concentrations results in cells cultured at a non-physiological concentration of approximately 18% oxygen. Neurons *in vivo* are exposed to much lower oxygen concentrations of between 0.55% and 8% (2, 3). Multiple studies have shown that cells cultured under more physiological oxygen conditions display features consistent with better health, including increased proliferation, plating efficiencies, viability, mitochondrial activity, and neurite outgrowth, as well as reduced senescence and chromosomal abnormalities (4). Accordingly, we cultured sympathetic neurons under more physiological oxygen concentrations, which enabled long-term survival and therefore latency and reactivation studies. Here, we directly compared latent infection under 5% versus atmospheric oxygen conditions to determine how changing oxygen levels impacts neuronal health and HSV-1 reactivation.

In preliminary studies examining infected sympathetic neurons, we observed that neurons appeared healthier in 5% oxygen (physioxic) versus atmospheric oxygen conditions, with brighter, rounder soma and less axonal fragmentation (Fig. 1A). To quantify these changes and exclude the complicating factor of infection, we cultured uninfected neurons isolated from the superior cervical ganglia (SCG) of newborn mice under physioxic or atmospheric conditions for 30 days and used a previously described matrix to assess neuronal health (5) (Tables 1 and 2; Fig. 1B and C). These matrices consider factors such as soma size, phase brightness, fragmentation, vesiculation, and blebbing, and scores and neuronal health are inversely related as higher scores represent poorer neuronal health. Neurons under physioxia demonstrated significantly reduced scores beginning at approximately 12–15 days post-plating. This differential in scores between physioxic and atmospheric conditions remained until the conclusion of the experiment.

Culturing neurons in microfluidic devices that isolate the axons from the soma compartment permits the specific treatment or infection of axons. However, this involves using miniscule amounts of media, leaving neurons more susceptible to hyperoxia under atmospheric conditions, and in our hands more susceptible to spontaneous degeneration. Therefore, we compared spontaneous degeneration of neurons cultured in microfluidic devices. At 10 days post-culture, we performed immunofluorescence using Beta III tubulin to visualize axons. Axons from neurons cultured under atmospheric conditions demonstrated significantly more fragmentation than those cultured under physioxic conditions (Fig. 1D). This was quantified using a degeneration index based on quantification of the ratio of fragmented axons over the total axon area (6) (Fig. 1E). Therefore, the health of neurons isolated from the SCG is improved following incubation under physiologic conditions versus atmospheric conditions independently of infection.

We next investigated the impact of physiological oxygen concentration on HSV-1 latency and reactivation. Neurons were cultured under atmospheric or physioxic conditions, and infected with HSV-1 Stayput-GFP in the presence of viral DNA replication inhibitor acyclovir (ACV) as previously described (1). At 6 days post-infection, ACV was washed out, and neurons were cultured for an additional 2 days. We used this 8-day post-infection time point as our latent time point. Latent cultures had equivalent levels of viral DNA copy numbers relative to host DNA (Fig. 2A), suggesting that the number of viral genomes capable of establishing latency remains similar despite

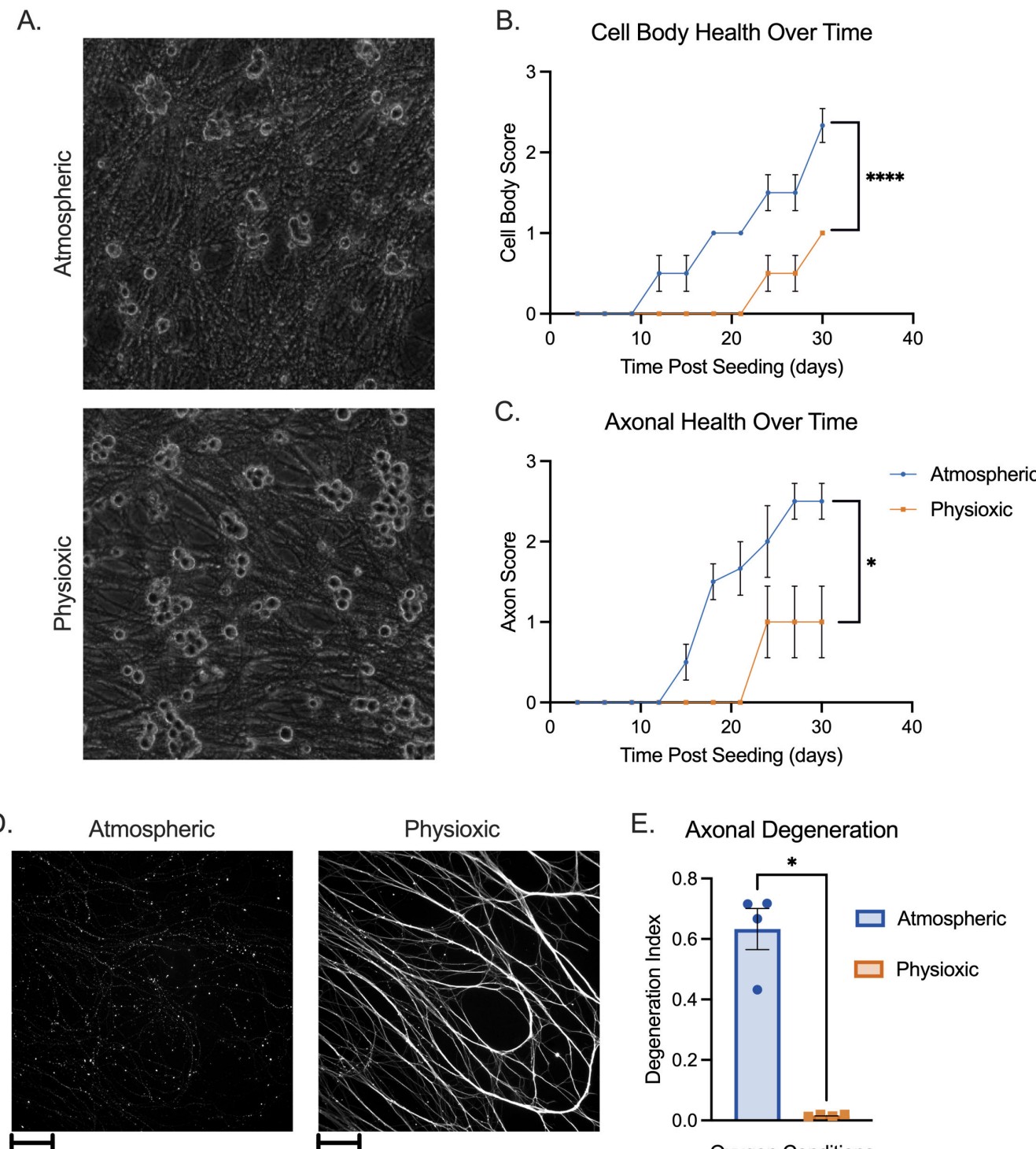

**FIG 1** Neuronal health in atmospheric versus physioxic incubation conditions. (A–E) Primary sympathetic neurons were incubated in atmospheric (approximately 18% oxygen) or physioxic (5% oxygen) conditions. (A) Neurons were latently infected with Stayput-GFP at an MOI of 10 PFU/cell in the presence of acyclovir (ACV; 50 µM) for 10 days and cultures were imaged using phase contrast to document soma morphology. (B and C) Neurons were incubated for up to 30 days post-plating. Scores representing cell body (B) and axon (C) health, as described in Tables 1 and 2, were recorded over time. Biological replicates from two independent dissections; statistical comparisons were made using two-way analysis of variance. (D and E) Neurons were plated in microfluidic chambers. Ten days post-plating, cultures were fixed and stained for neuronal marker Beta III tubulin (white) (D). Scale bar 100 µm. Axonal degeneration was analyzed using these images (E). Biological replicates from four independent experiments; statistical comparisons were made using unpaired non-normal t-test. Individual biological replicates along with the means and SEMs are represented. *P < 0.05; **P < 0.01.

**TABLE 1** Cell body scoring index

| Score | Description |
| --- | --- |
| 0 | Large, phase bright cell bodies. Clear with no fragmentation or vesiculation |
| 1 | Small, phase bright cell bodies. Clear with no fragmentation or vesiculation |
| 2 | Cell bodies do not have fragmentation but are not phase-bright. Sometimes appear transparent |
| 3 | Cell bodies with fragmentation but few dead neurons or corpses |
| 4 | Cell bodies with fragmentation with many corpses present and neurons starting to detach |
| 5 | Complete cell death. Neurons detached |

oxygen concentration, at least in this model system where ACV is used to promote latency establishment. Unlike most viral transcripts, one viral long noncoding RNA, known as the latency-associated transcript (LAT), is abundantly expressed during latency and can therefore be used to measure HSV latent infection (7). In parallel with latent viral DNA load, LAT expression was not statistically significantly different between the two conditions (Fig. 2B). Therefore, culturing primary neurons in physiological oxygen conditions did not impact the ability of HSV to establish a latent infection.

Following latency establishment and analysis, reactivation was induced by PI3-kinase inhibition using LY294002, a well-characterized trigger of HSV-1 reactivation which mimics the loss of a branch of the nerve growth factor (NGF) signaling pathway in neurons (8). HSV-1 reactivation proceeds in a two-step process. Phase I gene expression precedes "full reactivation" (or "Phase II") as a transcriptional burst of all classes of lytic viral genes (1, 9–12). Full reactivation is characterized by an ordered transcriptional cascade wherein viral immediate early gene transcription precedes and is essential to early gene expression, which is required for viral DNA replication, and subsequent late gene transcription. Ultimately, a new infectious virus is produced during full reactivation.

Phase I was interrogated through lytic gene expression analysis at 18 hours post-treatment. Phase I gene expression was similar between the two conditions (Fig. 2D through F), with a statistically nonsignificant trend toward lower viral transcription under physioxic conditions. Stayput-GFP has both a glycoprotein H (gH) deletion, which renders the virus deficient in cell-to-cell spread, and a GFP tag on late viral protein US11 (1). Therefore, Stayput-GFP permits the quantification of individual fully reactivating neurons. The number of Us11-GFP-positive neurons was quantified at 48 hours post-treatment with LY294002, which is indicative of Phase II reactivation. Interestingly, full reactivation was significantly reduced in physioxic versus atmospheric cultures (Fig. 2C). The average number of reactivating neurons was approximately 82 per well under atmospheric conditions versus 12 in physioxic conditions, demonstrating a sevenfold decrease under physioxia. Therefore, culturing neurons in more physioxic conditions results in equivalent latency and Phase I reactivation but reduced progression to full Phase II reactivation.

In our hands, the culturing of neurons under more physioxic conditions was necessary to improve neuronal health for longer-term latency studies. Physioxic conditions will also permit more consistent survival in microfluidic chambers, allowing long-term studies using axonal-specific treatments or infections. This report highlights the importance of accounting for oxygen concentration while studying the biology of primary neurons and microbial infections in all cell types. These findings also provide further mechanistic considerations regarding how neuronal signaling pathways mediate

**TABLE 2** Axon scoring index

| Score | Description |
| --- | --- |
| 0 | Axons are totally smooth with no blebbing or fragmentation. Branched and form a spider web-like network |
| 1 | Axons are smooth but grow straight |
| 2 | Blebbing on the axons but no apparent fragmentation |
| 3 | Fragmentation starting to appear in <50% of the neurons |
| 4 | Fragmentation in >50% of the neurons |
| 5 | No axons remaining |

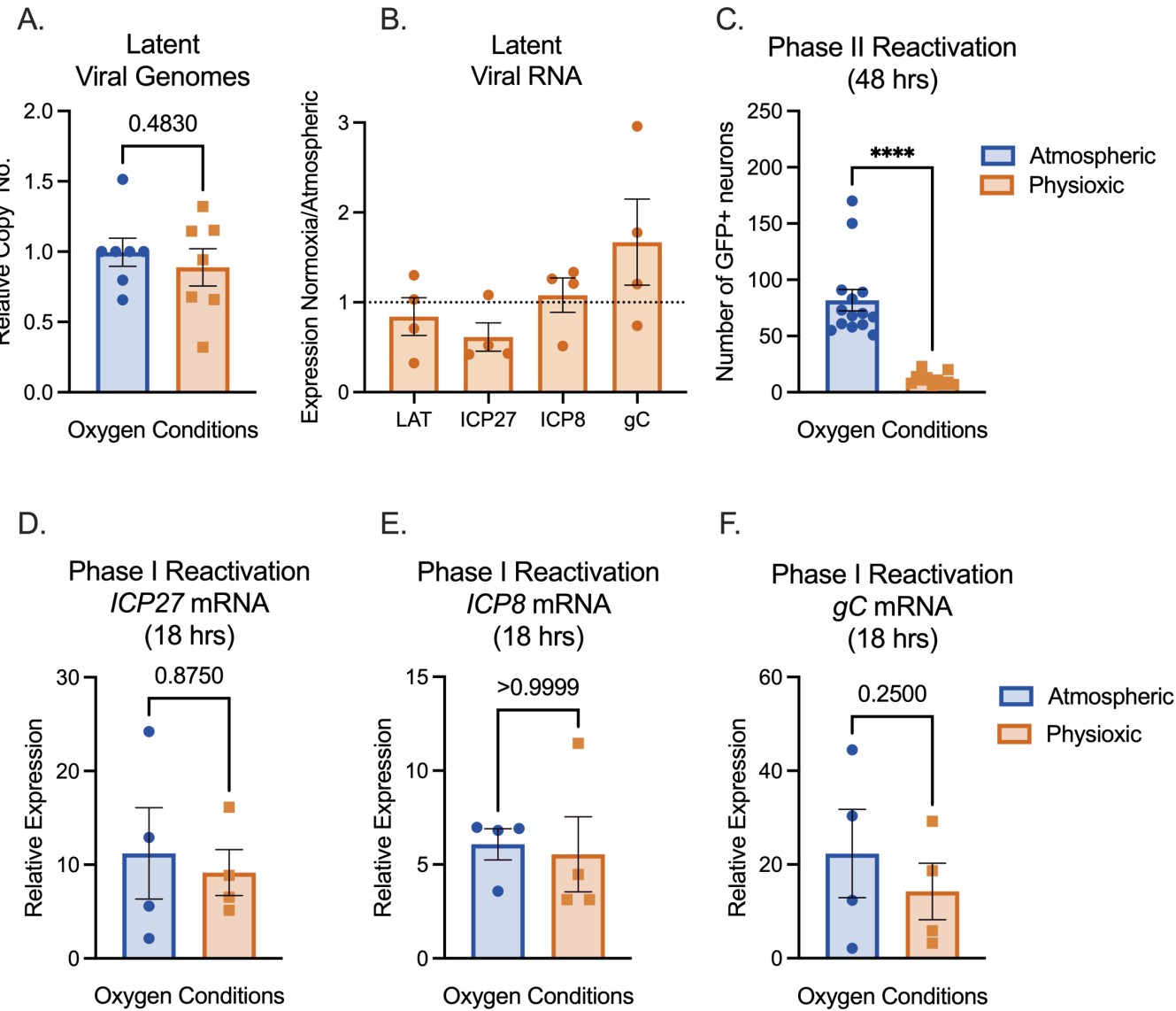

**FIG 2** Physioxic incubation conditions reduce HSV-1 reactivation. (A–F) Primary neurons were latently infected with Stayput-GFP at an MOI of 10 PFU/cell in the presence of acyclovir (ACV; 50 µM) for 6 days and then reactivated 2 days after the removal of acyclovir with LY294002 (20 µM). Quantification of relative latent viral DNA load (A) and LAT expression (B) at 8 days post-infection. (C) Quantification of the number of GFP-positive neurons at 48 hours post-stimulus. (D–F) Relative viral gene expression at 18 hours post-stimulus compared to latent samples quantified by reverse transcription-quantitative PCR (RT-qPCR) for immediate early viral gene ICP27 (D), early viral gene ICP8 (E), or late gene gC (F) normalized to cellular control mGAPDH. Biological replicates from four independent dissections; statistical comparisons were made using paired $t$-tests. Individual biological replicates along with the means and SEMs are represented. *$P < 0.05$; **$P < 0.01$.

HSV-1 reactivation from latency. Previously, the impact of hypoxia (1% oxygen), as an inhibitor of mRNA translation and therefore a neuronal stressor, on HSV-1 reactivation has been shown to be sufficient to trigger Phase II reactivation in culture models (13). Now, we conversely demonstrate that atmospheric, hyperoxic oxygen conditions are associated with greater neuronal stress and enhance Phase II reactivation triggered by LY294002. That both too little and too much oxygen in culture induces a state of stress and ultimately enhances HSV-1 reactivation is unsurprising. However, the signaling pathways through which oxygen alters reactivation require further investigation. Oxygen concentration is a balancing act in neurons; low oxygen concentrations are associated with the activation of transcription factor hypoxia-inducible factor, which regulates a number of processes in the nervous system including glycolysis, iron

metabolism, erythropoiesis, angiogenesis, and cell death (14). High oxygen concentrations are associated with reactive oxygen species production, which can trigger pathways involved in excitability, DNA damage, and inflammation within neurons (15–17). Further investigation into the role of these pathways in Phase I gene expression or full HSV-1 reactivation and under which environmental triggers they would be employed is required.

## Primary neuronal cultures

Sympathetic neurons from the SCG of post-natal day 0–2 (P0-P2) CD1 Mice (Charles River Laboratories) were dissected as previously described (18). Rodent handling and husbandry were carried out under animal protocols approved by the Animal Care and Use Committee of the University of Virginia (UVA). Ganglia were briefly kept in Leibovitz's L-15 media with 2.05 mM l-glutamine before dissociation in collagenase type IV (1 mg/mL) followed by trypsin (2.5 mg/mL) for 20 minutes; each dissociation step was at 37°C. Dissociated ganglia were triturated, and approximately 10,000 neurons per well were plated onto rat tail collagen in a 24-well plate. Sympathetic neurons were maintained in feeding media: Neurobasal Medium supplemented with PRIME-XV IS21 Neuronal Supplement (Irvine Scientific), 50 ng/mL Mouse NGF 2.5S (Alomone labs), 2 mM l-Glutamine, and 100 µg/mL Primocin (Invivogen). Aphidicolin (3.3 µg/mL) was added to the media for the first 5 days post-dissection to select against proliferating cells. Criteria for scoring and evaluating the health of neuronal cell bodies and axons are shown in Tables 1 and 2, respectively. Scoring was applied to entire fields of view (FOV) at 10× magnification, containing approximately ~200 neurons per FOV. In select experiments, neurons were plated in microfluidic chambers. Briefly, standard photolithography was used to fabricate channel array devices as described previously (19). Sylgard 184 (Dow Corning) was prepared according to the manufacturer's recommended procedure and poured into the mold. After curing at 95°C for at least 72 hours, individual polydimethylsiloxane chambers were cut out, sterilized in 70% EtOH, and placed upon glass coverslips coated with poly-D-lysine (100 µg/mL) and laminin (2 µg/mL).

## Preparation of HSV-1 virus stocks

Stocks of Stayput Us11-GFP (strain SC16) for *in vitro* experiments were propagated and titrated on gH-complementing F6 cells (5). Vero F6 cells were maintained in Dulbecco's modified Eagle's medium (Gibco) supplemented with 10% FetalPlex (Gemini Bio-Products) and 250 µg/mL of G418/Geneticin (Gibco).

## Establishment and reactivation of latent HSV-1 infection in primary neurons

P6-8 SCG neurons were infected with Stayput Us11-GFP at MOI 10 PFU/cell, (assuming 10,000 cells per well) in Dulbecco's Phosphate Buffered Saline (DPBS) + $CaCl_2$ + $MgCl_2$ supplemented with 1% fetal bovine serum, 4.5 g/L glucose, and 10 µM ACV for 3.5 hours at 37°C. The inoculum was replaced with feeding media (as described above) with 50 µM ACV. Six days post-infection, ACV was washed out and replaced with feeding media alone. Reactivation was reported by quantifying the numbers of GFP-positive neurons following the addition of 20 µM LY294002 (Tocris) or 60 µM forskolin (Tocris).

## Analysis of viral DNA load and mRNA expression by reverse transcription-quantitative PCR

To assess the relative expression of HSV-1 mRNA, total RNA was extracted from approximately 10,000 neurons using the Quick-RNA Miniprep Kit (Zymo Research) with an on-column DNase I digestion. mRNA was converted to cDNA using the Maxima First Strand cDNA Synthesis Kit for reverse transcription-quantitative PCR (RT-qPCR) (Fisher Scientific), using random hexamers for first-strand synthesis and equal amounts of RNA (20–30 ng/reaction). To assess viral DNA load, total DNA was extracted from approximately 10,000 neurons using the Quick-DNA Miniprep Plus Kit (Zymo Research). qPCR

was carried out using PowerUp SYBR Green Master Mix (ThermoFish Scientific). The relative mRNA or DNA copy number was determined using the comparative CT (ΔΔCT) method normalized to mRNA or DNA levels in latently infected samples. Viral RNAs were normalized to mouse reference gene mGAPDH RNA. All samples were run in triplicate on an Applied Biosystems QuantStudio 6 Flex Real-Time PCR System with the mean fold change compared to the calculated reference gene. The sequence of both forward and reverse primers used has been published previously (5).

## Immunofluorescence

Neurons were fixed for 15 minutes in 4% formaldehyde and blocked for 1 hour in 5% bovine serum albumin and 0.3% Triton X-100, and incubated overnight in primary antibody (Beta III Tubulin, EMD Millipore Cat #9354 at 1:500). Following primary antibody treatment, neurons were incubated for 1 hour in Alexa Fluor 488-conjugated secondary antibodies for multicolor imaging (Invitrogen). Nuclei were stained with Hoechst 33258 (Life Technologies). Images were acquired using an sCMOS charge-coupled device camera (pco.edge) mounted on a Nikon Eclipse Ti inverted epifluorescent microscope using NIS-Elements software (Nikon). Images were analyzed using ImageJ.

## Statistical analysis

Power analysis was used to determine the appropriate sample sizes for statistical analysis. All statistical analysis was performed using Prism V10. The normality of the data was determined with the Kolmogorov-Smirnov test. Specific analyses are included in the figure legends.

## ACKNOWLEDGMENTS

This work was supported by the National Institutes of Health grants NS105630 (A.R.C.), T32GM008136 (S.A.D.), and T32AI007046 (A.L.W.), and The Owens Family Foundation (A.R.C.).

## AUTHOR AFFILIATION

[1]Department of Microbiology, Immunology and Cancer Biology, University of Virginia, Charlottesville, Virginia, USA

## PRESENT ADDRESS

Sara A. Dochnal, Department of Anesthesiology, University of California, San Diego, La Jolla, California, USA

## AUTHOR ORCIDs

Sara A. Dochnal  http://orcid.org/0000-0003-3011-567X

## FUNDING

| Funder | Grant(s) | Author(s) |
| --- | --- | --- |
| HHS \| National Institutes of Health (NIH) | NS105630 | Anna R. Cliffe |
| HHS \| National Institutes of Health (NIH) | T32GM008136 | Sara A. Dochnal |
| HHS \| National Institutes of Health (NIH) | T32AI007046 | Abigail L. Whitford |
| Owens Family Foundation (The Owens Family Foundation) | | Anna R. Cliffe |

## AUTHOR CONTRIBUTIONS

Sara A. Dochnal, Conceptualization, Data curation, Formal analysis, Funding acquisition, Investigation, Methodology, Visualization, Writing – original draft, Writing – review and editing | Patryk A. Krakowiak, Data curation, Investigation, Validation, Visualization, Writing – review and editing | Abigail L. Whitford, Data curation, Investigation, Validation | Anna R. Cliffe, Funding acquisition, Methodology, Project administration, Resources, Writing – review and editing

## ADDITIONAL FILES

The following material is available online.

### Open Peer Review

**PEER REVIEW HISTORY (review-history.pdf).** An accounting of the reviewer comments and feedback.

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
