## [Reviewer comments · Microbiology Spectrum]

Microbiology Spectrum

Physiological oxygen concentration during sympathetic primary neuron culture improves neuronal health and reduces HSV-1 reactivation.

Sara Dochnal, Patryk Krakowiak, Abigail Whitford, and Anna Cliffe

Corresponding Author(s): Sara Dochnal, University of California at San Diego

Review Timeline:

Submission Date:	August 14, 2024
Editorial Decision:	September 11, 2024
Revision Received:	October 7, 2024
Accepted:	October 14, 2024

Editor: Donna Neumann

Reviewer(s): The reviewers have opted to remain anonymous.

Transaction Report:

DOI: <https://doi.org/10.1128/spectrum.02031-24>

Re: Spectrum02031-24 (Physiological oxygen concentration during sympathetic primary neuron culture improves neuronal health and reduces HSV-1 reactivation.)

Dear Dr. Sara Dochnal:

Thank you for the privilege of reviewing your work. Below you will find my comments, instructions from the Spectrum editorial office, and the reviewer comments.

The reviews were highly positive, but some additional details need to be included in the figures and the Discussion section should be edited according to reviewers comments.

Revision Guidelines

Sincerely,
Donna Neumann
Editor
Microbiology Spectrum

Reviewer #1 (Comments for the Author):

Studying biological processes in vitro is met with many challenges, including those that result in research conducted at less-than-ideal physiological conditions. Indeed, herpes simplex virus-1 (HSV-1), which undergoes latency in neurons, is challenging to study in such in vitro contexts due to the culturing conditions of standard cell culture incubators. This is particularly true for

oxygen levels. In vivo, neurons are exposed to low oxygen concentrations, ranging from ~0.55 - 8%. However, in standard tissue culture settings, neurons are subjected to a much higher, non-physiological concentration of ~18%. This could, in turn, impact the studies in these cells, imparting stress on the neurons, as well as hamper long-term studies, where the cell health is impacted. Such conditions could impact both HSV-1 latency and reactivation studies. Thus, in the current manuscript, the authors have evaluated the impact of culturing neurons at physioxic (5% oxygen) versus atmospheric oxygen on HSV-1 latency and reactivation. In sum, they find that while the neuronal cell health is improved with culturing at physioxic conditions, latency is established in neurons cultures at both physioxic and atmospheric conditions. However, the authors find physioxic conditions result in a reduction in HSV-1 reactivation. Nonetheless, the authors demonstrate the physioxic conditions are a viable means for long-term studies that require adequate neuronal cell health, which still retain the ability of HSV-1 to undergo latency and reactivation. The manuscript is presented in a logical fashion, and the authors have drawn the proper conclusions from their data. Additionally, the manuscript is well-written and easy to follow. Perhaps a very minor comment re: the text on lines 133-137 - the authors might reword this a bit so that it read a bit less repetitive. Overall, this work is solid and adds to our understanding of culture systems that support HSV-1 latency and reactivation.

Reviewer #2 (Comments for the Author):

In this manuscript, Dochnal et al evaluate how more physiological oxygen levels (5% physioxic O₂) compared with conventional atmospheric oxygen (18%) impacts the overall health of mouse primary sympathetic neuron cultures (isolated from the superior cervical ganglia of newborn mice) and their capacity to support HSV1 reactivation from latency in vitro. Culturing SCG neurons under physioxic O₂ resulted in healthier neurons with brighter, rounder soma and reduced axonal fragmentation. They found that the number of latent HSV1 genomes in neurons were similar in neurons maintained under physioxic O₂ vs atmospheric O₂. Similarly, HSV gene expression following inducible reactivation was similar in reactivation Phase 1, which involves transient genome-wide derepression of viral genes irrespective of their defined kinetic class. Remarkably, the authors report that progression of reactivation from Phase 1 to Phase 2, which is associated with viral DNA replication and infectious virus production, was reduced in neurons cultured under physioxic O₂ compared to atmospheric O₂. Taken together, the results of this study establish a new manner through which HSV reactivation from latency in a cultured neuron model is controlled by oxygen concentration. Overall, the manuscript is well written, the experiments are well executed, the results are significant, and the data are interesting, rigorous and compelling.

Specific points

1. The authors are not the first group to evaluate how O₂ concentration might impact HSV1 reactivation from latency. In particular, they should reference and acknowledge prior studies showing that O₂ concentration (hypoxia) regulates reactivation using a related cultured neuron model in the introduction (PMID: 22802527)
2. P5, lines 93. The description of scoring criteria is critical to understanding Fig1B/C and as such is not appropriate as supplemental content. This information needs to be incorporated into the main text.
3. Fig 1 B/C/E. Some indication of # neurons evaluated / # axons evaluated is required.
4. This study presents a significant discovery about the regulation of progression into Phase II that deserves substantially more attention in the Discussion section. The author's should also point out / incorporate prior published work (PMID: 22802527) into their revised discussion that better presents how oxygen concentration impacts reactivation from latency in cultured SCG neurons. Specifically, too little O₂ (hypoxia) triggers phase I/II reactivation, whereas now they show that too much O₂ (atmospheric) promotes progression of inducible reactivation through Phase II and physioxic O₂ allows phase I but restricts phase II progression. Some speculation regarding how oxygen sensing might tune reactivation is also warranted.

We thank the reviewers for their enthusiasm, consideration, and thorough feedback, which we address below, and we feel have enhanced the manuscript.

Reviewers' Comments to Authors:

Reviewer 1

“Perhaps a very minor comment re: the text on lines 133-137 - the authors might reword this a bit so that it read a bit less repetitive” (Reviewer 1)

We have shortened these lines (now lines 141-142) to eliminate redundancy.

Reviewer 2

“1. The authors are not the first group to evaluate how O₂ concentration might impact HSV1 reactivation from latency. In particular, they should reference and acknowledge prior studies showing that O₂ concentration (hypoxia) regulates reactivation using a related cultured neuron model in the introduction (PMID: 22802527)”

This reviewer is correct in that we are not the first group to investigate the impact of oxygen concentration on HSV-1 reactivation-nor do we claim to be. Our introduction highlights the importance of accounting for physiologically relevant oxygen levels in any model system and does not specifically aim to investigate the impact of oxygen signaling to HSV-1 reactivation. Importantly, long-term culture in more physiological oxygen levels should be taken into account when using hypoxia to induce cellular changes, as these experiments often involve shifts from hyperoxia to levels that are much closer to physioxia. Some lines have been re-worded to reflect this (line 117, lines 130-131).

“2. P5, lines 93. The description of scoring criteria is critical to understanding Fig1B/C and as such is not appropriate as supplemental content. This information needs to be incorporated into the main text.”

An additional description of factors the matrices consider is now included in the main text, along with explicit indication that score and neuronal health and negatively correlated (lines 93-96). Tables have been moved to the materials and methods section (lines 211- 214, lines 221-224) and are referenced in the Figure 1 Legend.

“3. Fig 1 B/C/E. Some indication of # neurons evaluated / # axons evaluated is required.”

The scoring matrices are based on the analysis of entire fields of view within wells, which contain approximately ~200 neurons each. We have now inserted this information within the materials and methods (lines 211-214).

4. This study presents a significant discovery about the regulation of progression into

Manuscript Spectrum02031-24
Response to Reviewers

Phase II that deserves substantially more attention in the Discussion section. The author's should also point out / incorporate prior published work (PMID: 22802527) into their revised discussion that better presents how oxygen concentration impacts reactivation from latency in cultured SCG neurons. Specifically, too little O₂ (hypoxia) triggers phase I/II reactivation, whereas now they show that too much O₂ (atmospheric) promotes progression of inducible reactivation through Phase II and physioxic O₂ allows phase I but restricts phase II progression. Some speculation regarding how oxygen sensing might tune reactivation is also warranted.

We have added the appropriate reference (lines 175) and a broader discussion on the impact of oxygen levels on reactivation from latency towards the conclusion of the main body of the text in lines 171-192.

Re: Spectrum02031-24R1 (Physiological oxygen concentration during sympathetic primary neuron culture improves neuronal health and reduces HSV-1 reactivation.)

Dear Dr. Sara Dochnal:

Your manuscript has been accepted, and I am forwarding it to the ASM production staff for publication. Your paper will first be checked to make sure all elements meet the technical requirements. ASM staff will contact you if anything needs to be revised before copyediting and production can begin. Otherwise, you will be notified when your proofs are ready to be viewed.

Sincerely,
Donna Neumann
Editor
Microbiology Spectrum